

# Origin and evolution of GATA2a and GATA2b in teleosts: insights from tongue sole, *Cynoglossus semilaevis*

Jinxiang Liu, Jiajun Jiang, Zhongkai Wang, Yan He and Quanqi Zhang

Key Laboratory of Marine Genetics and Breeding, Ministry of Education, Ocean University of China, Qingdao, China

## ABSTRACT

**Background.** Following the two rounds of whole-genome duplication that occurred during deuterostome evolution, a third genome duplication occurred in the lineage of teleost fish and is considered to be responsible for much of the biological diversification within the lineage. GATA2, a member of GATA family of transcription factors, is an important regulator of gene expression in hematopoietic cell in mammals, yet the role of this gene or its putative paralogs in ray-finned fishes remains relatively unknown.

**Methods.** In this study, we attempted to identify GATA2 sequences from the transcriptomes and genomes of multiple teleosts using the bioinformatic tools MrBayes, MEME, and PAML. Following identification, comparative analysis of genome structure, molecular evolution rate, and expression by real-time qPCR were used to predict functional divergence of GATA2 paralogs and their relative transcription in organs of female and male tongue soles (*Cynoglossus semilaevis*).

**Results.** Two teleost GATA2 genes were identified in the transcriptomes of tongue sole and Japanese flounder (*Paralichthysolivaceus*). Synteny and phylogenetic analysis confirmed that the two genes likely originated from the teleost-specific genome duplication . Additionally, selection pressure analysis predicted these gene duplicates to have undergone purifying selection and possible divergent new functions. This was supported by differential expression pattern of GATA2a and GATA2b observed in organs of female and male tongue soles.

**Discussion.** Our results indicate that two GATA2 genes originating from the first teleost-specific genome duplication have remained transcriptionally active in some fish species and have likely undergone neofunctionalization. This knowledge provides novel insights into the evolution of the teleost GATA2 genes and constituted important groundwork for further research on the GATA gene family.

**Submitted** 16 October 2015
**Accepted** 20 February 2016
**Published** 21 March 2016

Corresponding author
Quanqi Zhang, qzhang@ouc.edu.cn

## INTRODUCTION

GATA transcription factors are evolutionarily conserved proteins that bind the consensus motif WGATAR in gene regulatory regions (*Evans, Reitman & Felsenfeld, 1988*; *Whitelaw et al., 1990*). GATA proteins are characterized by the conserved N-terminal and C-terminal zinc finger motifs. The N-terminal zinc finger is required for DNA binding, whereas the C-terminal zinc finger stabilizes binding and physical interaction with other co-factors

(*Yang & Evans, 1992*). All GATA proteins are essential to animal developmental processes, including germ layer specification, hematopoiesis, and cardiogenesis (*Sorrentino, Gajewski & Schulz, 2005*). All the GATA family members can induce reprogramming and substitute for *Oct4* (*Shu et al., 2015*).

GATA has been identified in vertebrates, invertebrates, fungi, and plants (*Lowry & Atchley, 2000*), as well as protostomes and deuterostomes (*Patient & McGhee, 2002*). The GATA gene family, including GATA123 and GATA456 subfamilies (*Gillis, Bowerman & Schneider, 2008*), has undergone significant expansion after whole-genome duplication in vertebrate lineages. To date, two GATA genes have been identified in the sea urchin *Strongylocentrotus purpuratus*, and two in the hemichordate *Saccoglossus kowalevskii*, the urochordate *Ciona intestinalis*, and the cephalochordate *Branchiostoma floridae.* Meanwhile, six GATA transcription factors (GATA1 to GATA6) have been found in tetrapods and teleosts (*Gillis et al., 2009*).

Previous studies have verified multiple rounds of whole-genome duplication in vertebrate lineages, which may play a significant role in vertebrate evolution (*Hoegg & Meyer, 2005*; *Hoffmann, Opazo & Storz, 2012*; *Hughes, 1999*). Interestingly, a third whole-genome duplication event (3R) occurred in teleosts (*Amores et al., 1998*; *Postlethwait et al., 1998*). Teleost-specific genome duplication (TGD) provided more gene copies, contributing to the evolutionary and phenotypic diversification of teleosts. TGD-derived gene duplicates supported the cause–effect relationship between gene copy number and species diversity (*Siegel et al., 2007*). The duplicated genes might possess great divergence from their ancestors, as demonstrated by the changes in evolutionary rates, expression patterns, and regulatory mechanisms observed across the teleost lineage (*Braasch, Salzburger & Meyer, 2006*; *Hoegg & Meyer, 2007*; *Mulley, Chiu & Holland, 2006*). Duplicated genes have three main fates, that is, nonfunctionalization, subfunctionalization, and neofunctionalization (*Force et al., 1999*).

In teleosts, research investigating GATA2 has been minimal. To better understand the origination and functional divergence of GATA2 in teleost, this study aimed to investigate GATA2 gene(s) from the transcriptome of tongue sole, Japanese flounder, and other teleosts. Following identification of two GATA2 genes in the tongue sole, chromosomal synteny and phylogenetic analysis of these genes was performed to investigate the origin and evolution of GATA2 in teleosts. Then, analysis of genomic structure, molecular positive selection, and expression pattern of the two GATA2 genes in tongue sole were performed to identify potential changes in functionality for the duplicated GATA2 genes within the teleost lineage. This study provides evidence to support the GATA family expansion theory that the increase of GATA members follows the whole-genome duplication. It also lays the foundation for further evolutionary and functional studies of the GATA gene family in teleosts.

## MATERIALS AND METHODS

### Ethics statement

All research was conducted in accordance with the Institutional Animal Care and Use Committee of the Ocean University of China and with the China Government Principles

for the Utilization and Care of Vertebrate Animals Used in Testing, Research, and Training (State science and technology commission of the People's Republic of China for No. 2, October 31, 1988: http://www.gov.cn/gongbao/content/2011/content_1860757.htm).

## Fish

Healthy tongue sole (three females and three males) aged one year were chosen from a larger cohort population. The fish were anesthetized (MS-222 at 30 μg/mL) and then killed by severing spinal cord. Brain, heart, intestine, kidney, liver, spleen, and gonad organs were collected in triplicate from each fish. All of the samples were immediately frozen using liquid nitrogen and stored at −80 °C for total RNA extraction.

## Identification of GATA gene family sequences in the tongue sole

GATA gene family members were identified from Amazon molly (*Poecilia formosa*), fugu (*Takifugu rubripes*), medaka (*Oryzias latipes*), stickleback (*Gasterosteus aculeatus*), tetraodon (*Tetraodon nigroviridis*), and tilapia (*Oreochromis niloticus*) whose genomes are completely sequenced and available from the Ensembl database. The retrieved sequences were used as query sequences in BLAST searches. The mRNA sequences of GATA genes were identified using tBLASTn analysis from the tongue sole transcriptome previously sequenced by our laboratory. The transcriptome was generated from a total of 749,954 reads using a single 454 sequencing run and assembled into 62,632 contigs, of which 26,589 sequences were successfully annotated (*Wang et al., 2014b*). These fragments were used to search for the corresponding chromosomal regions containing in the tongue sole genome from NCBI (GenBank accession: PRJNA73987). *Cs*GATA2a was found on scaffold385_11, and *Cs*GATA2b was identified on scaffold57_8.

## Identification of GATA gene family sequences in the Japanese flounder

The sequences retrieved from tongue sole and the other six teleosts listed above were used as query sequences to search for *Po*GATA genes. The sequences were identified from the Japanese flounder transcriptome through tBLASTn analysis (*Wang et al., 2014a*). An unpublished Japanese flounder genome was used to search for the DNA sequences of *Po*GATA2a and *Po*GATA2b (Supplemental Information 1).

## GATA2 sequence alignment and phylogenetic analysis

The sequence alignments of GATA2a and GATA2b were based on their predicted peptide sequences using Clustal X with default parameters (*Chenna et al., 2003*). Phylogenetic trees were constructed to confirm the ortholog and paralog relationships of both duplicates. The sequences used to construct gene trees were retrieved from Ensembl and NCBI (species names, gene names, and accession numbers are available in Table S1). The most appropriate substitution model of molecular evolution was determined using JModelTest v2.1.4 (*Darriba et al., 2012*). To confirm the tree topologies, a Bayesian tree and a maximum likelihood tree were respectively constructed using MrBayes v3.2.2 (*Huelsenbeck & Ronquist, 2001*; *Ronquist et al., 2012*) and phyML v3.1 (*Guindon et al., 2010*). MrBayes was run for 400,000 generations with two runs and four chains in parallel and a burn-in of

25%. PhyML was run for 1,000 replications. Other parameters were based on the result of JModelTest.

## Tests for positive selection in GATA2a and GATA2b

A Bayesian tree was constructed using MrBayes based on GATA2a and GATA2b. The tree includes all species used for positive selection analyses (Table S1). The TIM3 + I + G model with base frequencies and substitution rate matrix estimated from the parameters (as suggested by JModelTest) was used. The standard site model in CODEML of PAML v4.7 was used to calculate selection pressures (*Yang, 2007*). The site model employed ML estimation of the ratio of nonsynonymous to synonymous substitutions ($d_N/d_S = \omega$) and nested likelihood ratio tests (LRTs) on a phylogeny tree.

## Genomic structure, motif, and synteny analysis of teleost GATA2 paralogs

Diagrams of exon–intron structures were obtained using the online Gene Structure Display Server 2.0 (GSDS: http://gsds.cbi.pku.edu.cn) with CDS and genomic sequences (*Hu et al., 2015*). Motifs in the candidate GATA2 DNA sequences were identified using MEME (*Bailey et al., 2009*). The Synteny Database (*Catchen, Conery & Postlethwait, 2009*) was used to generate dotplots of the human GATA2 gene region on chromosome Hsa3 and the genome of zebrafish to analyze the syntenic conservation between fish and human chromosomes.

## RNA isolation, cDNA synthesis, and qRT-PCR

Total RNA was extracted from organ samples with Trizol reagent (Invitrogen, Carlsbad, CA, USA) in accordance with the manufacturer's instructions. DNA was removed using DNase I (TaKaRa, Dalian, China) treated with 2 h at 37 °C, and the protein was digested using an RNAclean RNA Kit (Biomed, Beijing, China). The quality and quantity of the extracted RNA were identified via electrophoresis and Nanophotometer® Pearl (Implen GmbH, Munich, Germany). Frist-strand cDNA was synthesized using the PrimeScript™ RT-PCR Kit (TaKaRa) in accordance with the manufacturer's instructions.

Quantitative Real-time was conducted on a LightCycler 480 (Roche, Forrentrasse, Switzerland). The respective primer pairs for GATA2a and GATA2b were Cs-GATA2a-RT and Cs-GATA2b-RT (Table S2), which were designed by IDT (http://www.idtdna.com/Primerquest/Home/Index) in the 3′ UTR of both genes. Standard curves were established from a serial dilution of plasmids containing GATA2a, GATA2b, and reference gene RPL17 fragments. Efficiency values (91.58%, 88.25%, and 92.10%, respectively) were calculated by standard curves (*Boyle, Dallaire & MacKay, 2009*). cDNA from three females and males were diluted as templates (10 ng/μL) for sample assessment. The SYBR Green master mix (Roche, Basel, Switzerland) was used as the PCR detection system. Three male and three female individuals were collected. The same organ from three male or three female individuals were pooled as one sample for expression analysis, and the experiments for each pooled sample were performed in triplicate. Thermocycling consisted of an initial polymerase activation of 30 s at 94 °C, followed by 40 cycles at 94 °C for 15 s and 60 °C for 45 s. Product specificity was ensured through melting curve analysis which consisted of

40 cycles. RPL17 of tongue sole was used as the reference gene to normalize the expression which has been shown to be stably expressed between male and female tongue soles in multiple organ types (*Liu et al., 2014*). The sizes of GATA2a, GATA2b, and RPL17 amplicons were 121 bp, 122 bp, and 114 bp, and melt curve starting temperatures were 60 °C, 61 °C, and 60 °C, respectively. Data were analyzed through the $2^{-\Delta\Delta Ct}$ method.

## Statistical analysis

qRT-PCR data were statistically analyzed using one-way ANOVA on log10-transformed data followed by LSD test using SPSS 20.0, and $P < 0.05$ was considered to indicate statistical significance. All data were expressed as mean $\pm$ standard error of the mean (SEM).

## RESULTS

### Identification of GATA genes

We identified GATA1, GATA2a, GATA2b, GATA3, GATA4, GATA5, and GATA6 from the transcriptomes of tongue sole and Japanese flounder via tBLAST to infer the origin and evolutionary history of the GATA gene family in teleosts. Other GATA genes were searched from Ensembl and NCBI. The GATA family can be divided into the GATA123 and GATA456 subfamilies. Protein analysis showed that the GATA family in teleosts comprised two conserved zinc finger motifs at the N-terminal and the C-terminal domains. However, the different GATA paralogs in teleosts had varied lengths. Seven GATA genes, including two GATA2 genes (GATA2a and GATA2b), were detected in the teleost GATA family. Only six GATA genes were detected in tetrapods.

### Phylogenetic relationships and evolution of the GATA gene family

The identified DNA sequences were analyzed to investigate the evolutionary relationship of GATA genes among various teleosts using multiple sequence alignment with Clustal X. A phylogenetic tree of the GATA gene family was constructed using MrBayes and phyML based on the alignment results. The two programs inferred similar topologies, which indicated that the GATA gene family could be divided into seven well-conserved clades and two subfamilies in teleosts (Fig. 1).

Our results also indicated distinct ancestral relationship within each subfamily of the GATA gene family. A close relationship was observed between GATA2a/b and GATA3 within the GATA123 subfamily, and between GATA5 and GATA6 within the GATA456 subfamily.

### Phylogenetic analysis of teleost-specific GATA2a and GATA2b

Multiple amino acid alignment was conducted to explore the origin, generation, and differentiation of GATA2a and GATA2b in teleosts. The sequence similarity between GATA2a and GATA2b was 82.55%, with two highly conserved zinc finger motifs. Sequence alignment suggested the occurrence of two GATA2b-specific mutations in the N-terminal and C-terminal zinc fingers. Specifically, serine was dehydroxymethyled into glycine in the N-terminal zinc finger motif, and alanine was demethylated into glycine in the C-terminal

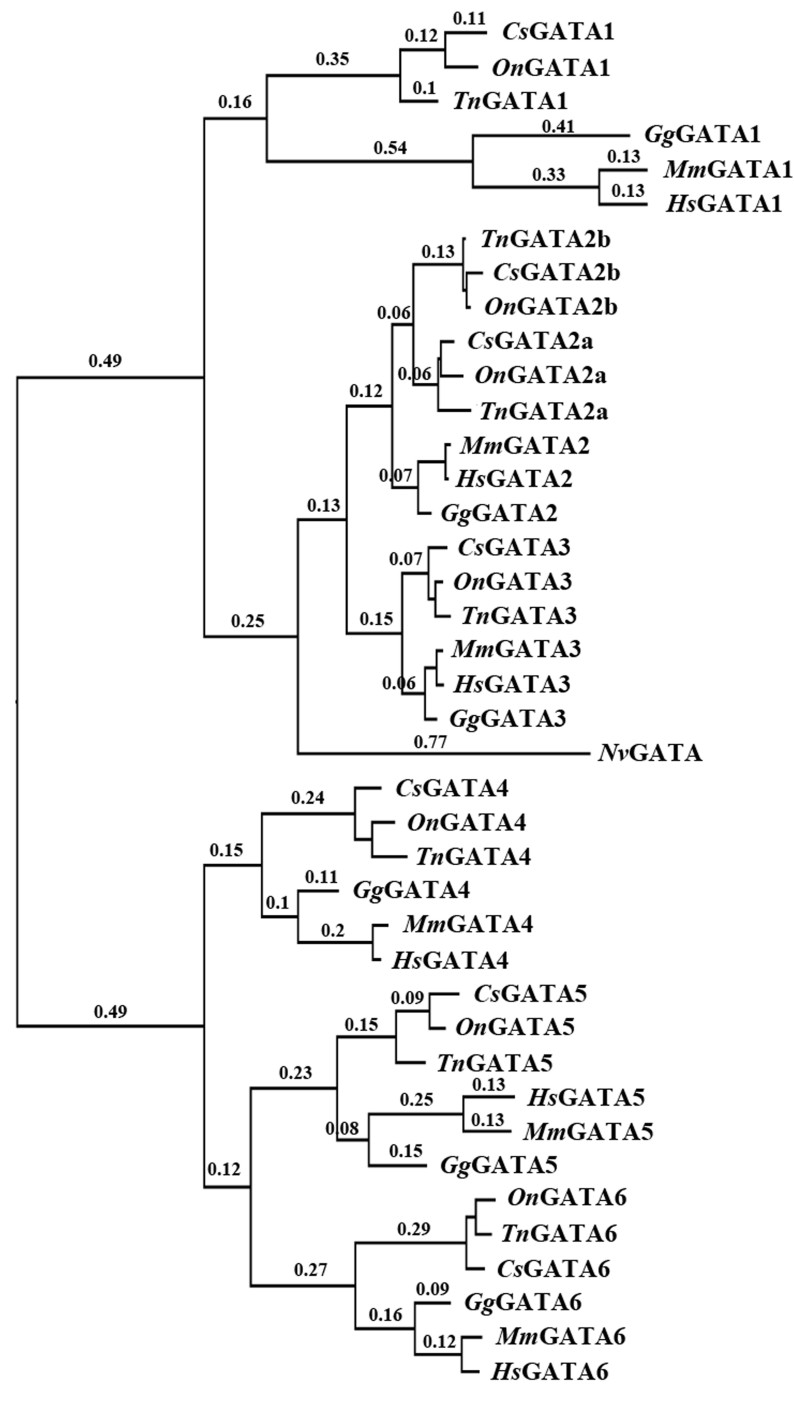

**Figure 1 Phylogenetic analyses of vertebrate GATA gene family.** Phylogenetic tree constructed using MrBayes with the TPM1uf + I + G model; MCMC = 400,000 generations. *On, Oreochromis niloticus; Cs, Cynoglossus semilaevis; Tn, Tetraodon nigroviridis; Mm, Mus musculus; Gg, Gallus gallus; Hs, Homo sapiens; Nv, Nematostella vectensis.*

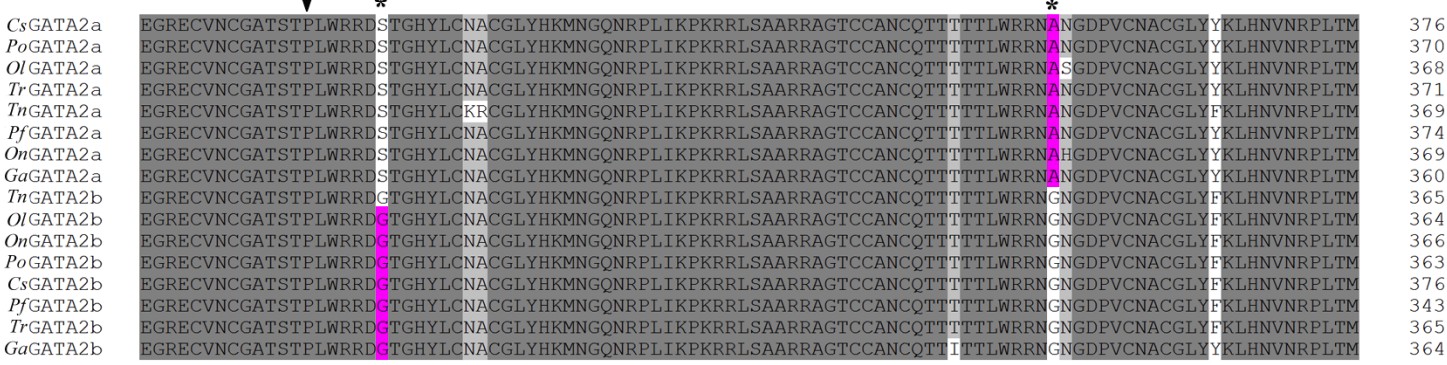

**Figure 2** **Partial multiple sequence alignment of the deduced GATA2a and GATA2b protein sequences.** In teleosts, two conserved zinc finger motifs were found in GATA2a and GATA2b (underlined sequences). Two GATA2b-specific mutations were identified in zinc finger motifs (star-shaped site). The proline site (arrowhead) in the N-terminal zinc finger motif had undergone positive selection.

zinc finger motif (Fig. 2 and Fig. S1). The N-terminal zinc finger motif can stabilize the binding and physically interact with other co-factors, and the C-terminal zinc finger motif is required for DNA binding. Thus the dehydroxymethylation and demethylation mutations might trigger protein structure alteration, and further affect the molecular functions in biological processes.

The phylogenetic trees of GATA2a and GATA2b in teleosts were constructed using MrBayes and phyML, with the human GATA2 sequence as an outgroup. The two trees were similar in topology with minimal bootstrap differences. Results indicated that teleost GATA2 genes could be divided into two well-conserved clusters: GATA2a and GATA2b (Fig. 3), implying that GATA2a and GATA2b in teleosts were probably generated from the same ancestor.

## Genomic structures of teleost GATA2

Gene structure graphics were constructed by the online program Gene Structure Display Server to analyze the evolutionary mechanism of GATA2. The graphics showed that both GATA2a and GATA2b had five exons in CDS, except for *Pf* GATA2b, which had an extra intron dividing the second exon into two segments. The lengths of each corresponding exon were highly conserved, but intron lengths varied among species. The GATA2a and GATA2b in fugu and tetraodon had the shortest intron lengths. In most teleosts, the GATA2b gene was longer than the GATA2a gene, which might infer that the two subtypes of GATA2 had undergone gene differentiation, that is, they originated from a common ancestor but diverged into two genes differing in protein structure and functions (Fig. S2A). This inference was further supported by motif prediction on teleost GATA2 by MEME. Four main motifs (motifs 1, 2, 3, and 4) were predicted in both GATA2a and GATA2b. An additional motif 4 was predicted at the end of GATA2b in most teleosts (Fig. S2B).

## Synteny analysis of teleost GATA2 paralogs

Chromosomal synteny analysis was carried out between human and zebrafish to test whether that GATA2 paralogs originated from whole-genome duplication. Conserved

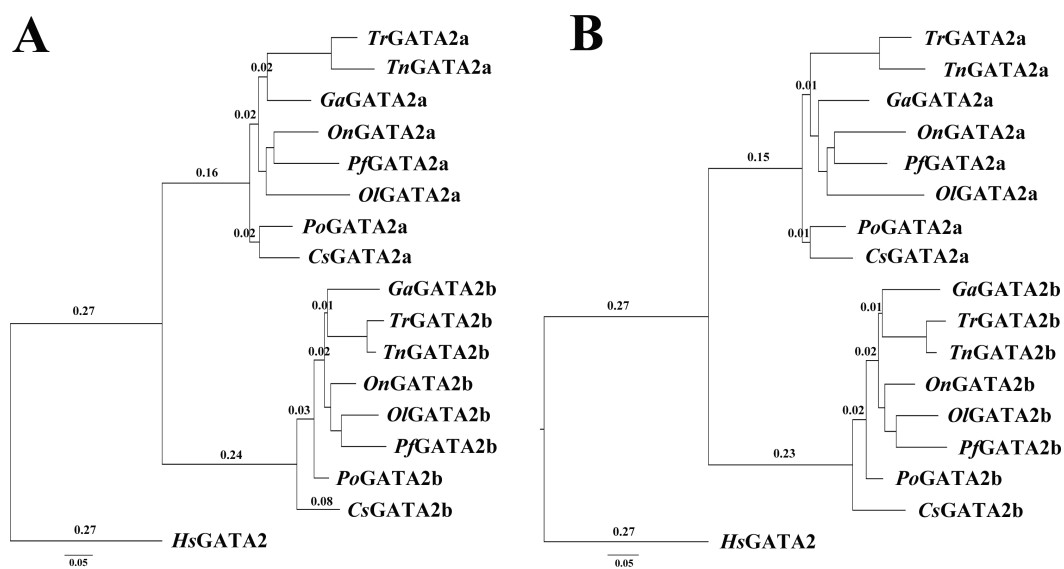

**Figure 3** **Phylogenetic analysis of teleost GATA2.** (A) Phylogenetic tree constructed based on GATA2a and GATA2b in teleosts by using MrBayes with the TIM3 + I + G model; MCMC = 400,000. (B) Maximum likelihood phylogenetic tree constructed using phyML with the TIM3 + I + G model. PhyML was run for 200 replications. *Po-P.olivaceus, Tr-T. rubripes, Pf-P. formosa, Ga-G. aculeatus.*

synteny dotplots showed that the GATA2a and GATA2b regions in zebrafish shared conserved synteny. The zebrafish GATA2a region on Dre11 shared conserved synteny neither with the zebrafish GATA2b region Dre6 nor with the human GATA2 region Hsa3 (Fig. 4). Previous studies confirmed that these chromosomes originated from the common ancestral chromosome and duplicated during the teleost-specific genome duplication (*Kasahara et al., 2007*; *Nakatani et al., 2007*).

Gene neighborhood analysis showed highly conserved synteny within GATA2a or GATA2b and between the two genes. In teleosts, the genes near GATA2a, except for some genes lost in tetraodon duplication era, were mostly conserved and shared the same direction (Fig. 5A). Long fragments consisting of several genes were lost in the upstream and downstream regions of GATA2b in Amazon molly and fugu, but the other genes remained conserved. Comparison of the upstream genes of GATA2a and GATA2b revealed that a fragment including four genes was conserved, albeit in opposite directions (indicated by blank pentagons) (Fig. 5B). A gene in the upstream region and a two-gene string in the downstream region (indicated by blank pentagons) were also highly conserved between GATA2a and GATA2b. These results implied that the genes neighboring teleost GATA2a or GATA2b were highly conserved, and more conserved among teleosts after duplication.

## Molecular evolution of teleost GATA2a and GATA2b

In general, phenotypic differences can arise from mutations affecting protein functions or changes in gene regulation (*Stainier et al., 1996*). Therefore, we examined the coding sequence evolution in two GATA2 paralogs to test for positive selection and potential functional changes in teleosts. The site models in PAML were used to assess different selective pressures. The estimation of positive selection based on the phylogenetic trees is

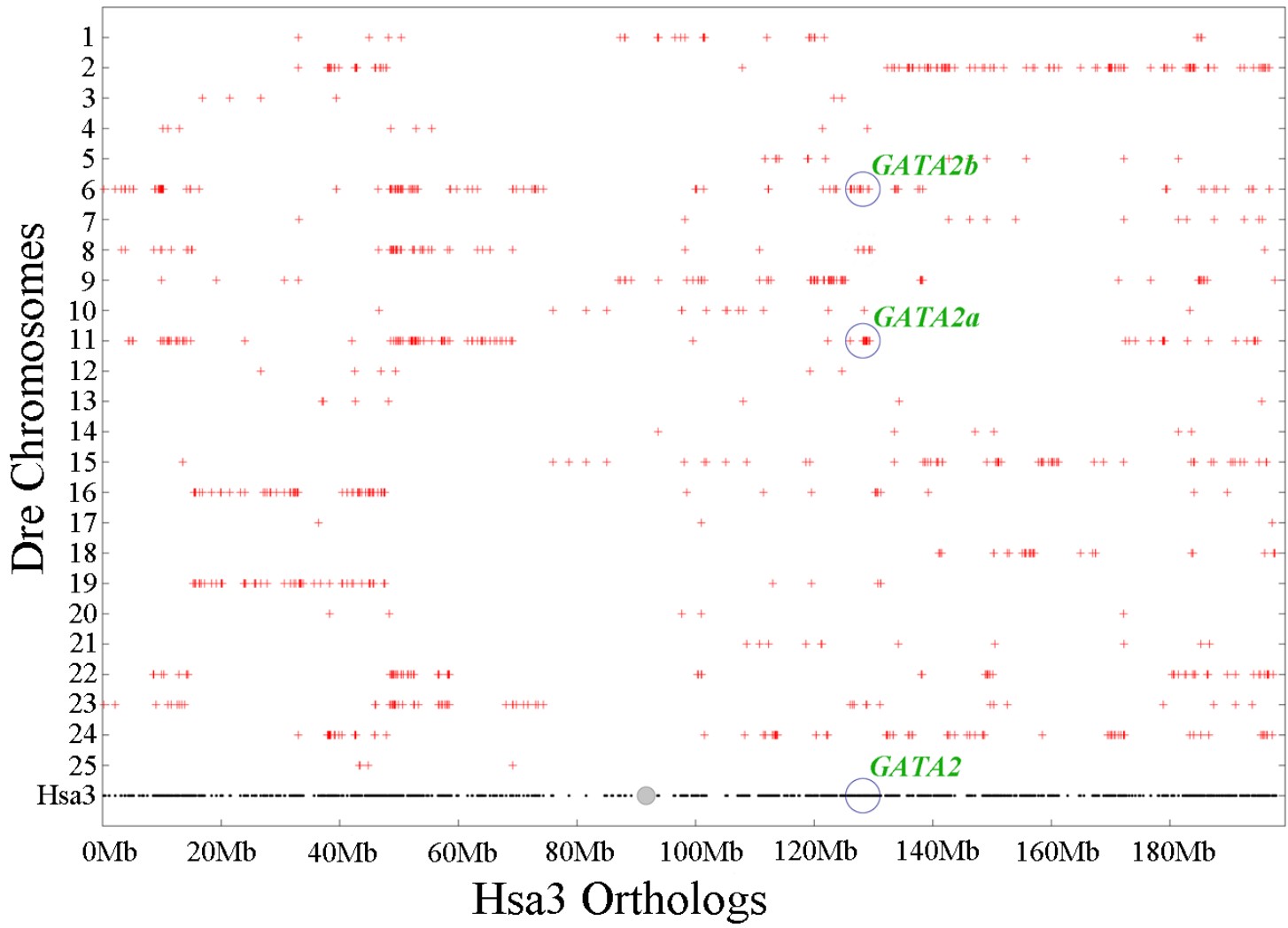

**Figure 4 Chromosome synteny analysis of teleost GATA2 paralogs.** Dotplots of the human GATA2 gene region on human chr3 show double conserved synteny to the two GATA2 paralogs in zebrafish on chromosomes Dre6 (GATA2b) and Dre11 (GATA2a).

shown in Fig. S3. Three model pairs (M0/M3, M1a/M2a, and M7/M8) were selected and compared with the site-specific codeml model to test whether variable $\omega$ ratios occurred at amino acid sites. The parameters and the LRT results are listed in Table 1.

In GATA2, M3 (discrete) was significantly better than M0 (one-ratio) ($P < 0.05$). Thus, M0 was rejected, indicating the extreme variation in selection pressure among amino acid sites. Overall, the GATA2 sequences had undergone positive selection. Additional tests with M1a (neutral) and M2a (selection), M7 (beta)/M8 (beta & $\omega$) and M8a were conducted using the chi2 program in PAML. The LRT significantly differed in the M7/M8 pair of GATA2b ($P < 0.05$). One candidate amino acid site for positive selection (356P, $P < 0.05$) was identified (356P*) through the Bayes Empirical Bayes (BEB) method of M8. No site under positive selection was identified in GATA2a. Then, the relationship between amino acid sites under positive selection and function divergence was analyzed. The site 356P with a posterior probability >0.95 was located in the C-terminal zinc finger in GATA2b,

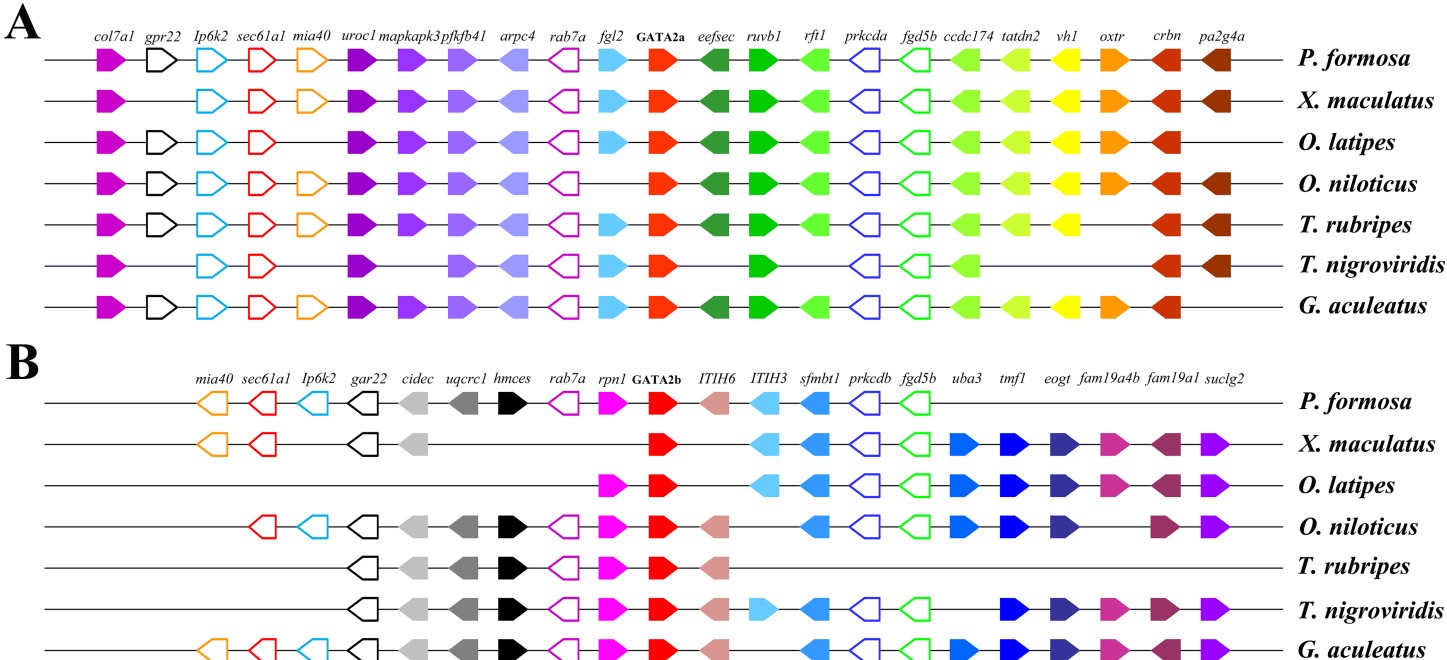

**Figure 5** **Chromosomal segments showing the conserved syntenic blocks containing GATA2a and GATA2b in teleosts.** The genes are represented by colored pentagons, and the gene names are indicated on top. Color pentagons indicate the same gene in different species and its respective genomic position in relation to several other genes. The pentagon's direction indicates the gene direction compared with the reference gene. The empty spaces indicate a region with other genes or the absence of the gene in the genome. The blank pentagons indicate conserved genes between GATA2a and GATA2b.

indicating that GATA2b, especially its motif, had experienced a strong selective pressure, which might develop mechanism adapting to water environment.

## Expression levels of GATA2a and GATA2b in organ

Quantitative real-time PCR using RNA extracted from multiple tongue sole organs was performed to test if transcription regulation of GATA2a and GATA2b had undergone divergence in teleosts. Both genes were expressed in all organs tested but possessed distinct levels of expression. Heart and the brain showed higher relative expression of GATA2a/b than other somatic organs in both sexes, and extraordinarily high GATA2b expression was found in the heart (Fig. S4). A sexual dimorphic expression pattern was observed in the gonads. In the ovary, GATA2a expression was hardly observed and GATA2b expression was very low (Fig. 6A), while in the testis, GATA2a expression was moderate and GATA2b expression was relatively high (Fig. 6B).

## DISCUSSION

### Expansion of vertebrate GATA transcription factor genes during multiple whole-genome duplications

GATA transcription factors play crucial roles in regulating the development and differentiation processes including hematopoiesis, cardiogenesis, and germ layer specification (*Holtzinger & Evans, 2005*; *LaVoie, 2003*). In the present study, seven GATA

**Table 1** Results of sites model analyses on the teleost GATA2 Bayesian gene tree.

| Tree | Model | lnL | κ | Null | LRT | df | P-value | site | BEB |
|---|---|---|---|---|---|---|---|---|---|
| GATA2a | M0 | −4832.177 | 2.463 | NA | | | | | |
| | M1a | −4775.943 | 2.642 | NA | | | | | |
| | M2a | −4775.943 | 2.642 | M1a | 0 | 2 | 1.000 | | |
| | M3 | −4732.818 | 2.523 | M0 | 198.718 | 4 | 0.000 | | |
| | M7 | −4733.231 | 2.524 | NA | | | | | |
| | M8a | −4735.152 | 2.540 | NA | | | | | |
| | M8 | −4733.231 | 2.524 | M7 | 0 | 2 | 1.000 | | |
| | | | | M8a | 3.842 | 1 | 0.050 | | |
| GATA2b | M0 | −4083.986 | 2.472 | NA | | | | | |
| | M1a | −4053.831 | 2.560 | NA | | | | | |
| | M2a | −4053.831 | 2.560 | M1a | 0 | 2 | 1.000 | | |
| | M3 | −4034.159 | 2.488 | M0 | 99.654 | 4 | 0.000 | | |
| | M7 | −4034.060 | 2.485 | NA | | | | | |
| | M8a | −4037.538 | 2.506 | NA | | | | | |
| | M8 | −4030.944 | 2.490 | M7 | 6.232 | 2 | **0.044** | 356(P) | 0.95 |
| | | | | M8a | 13.188 | 1 | **0.00028** | | |

**Notes.**

Abbreviations: lnL, *ln* likelihood; κ, Transition/transversion ratio; df, Degrees of freedom; NA, Not applicable.

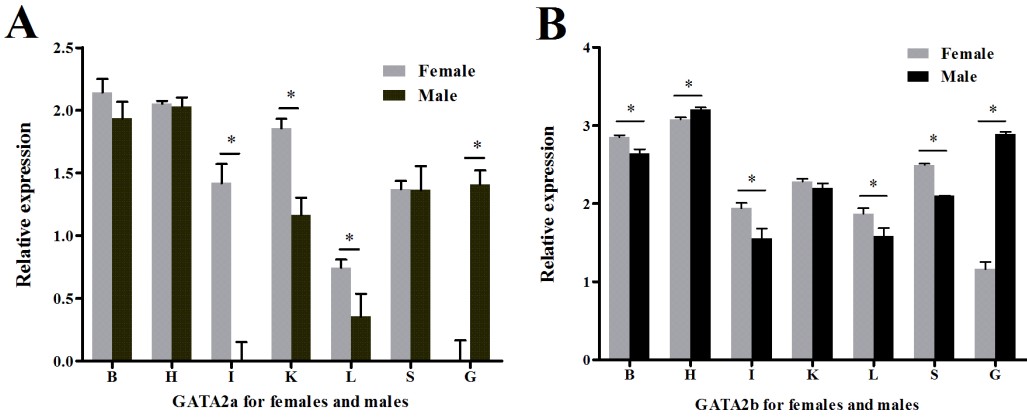

**Figure 6** **Relative expression levels of GATA2a and GATA2b in tongue sole tissues.** (A) Relative expression level of GATA2a in males and females. (B) Relative expression level of GATA2b in males and females. B, brain; H, heart; I, intestine; K, kidney; L, liver; O, ovary; T, testis; S, spleen.

genes were identified from both tongue sole and Japanese flounder transcriptomes. Indeed, all teleosts analyzed in this study possessed seven GATA genes, including six GATA genes shared with tetrapods and an additional teleost-specific GATA2 duplication. As teleosts have undergone a unique 3R genome duplication, some gene families became larger in teleosts than in tetrapods or chondrichthyes. Thus, the present results are consistent with previous reports that GATA gene family expansion occurred through genome duplication and that clade-specific conserved losses of duplicated paralogs occurred after duplication (*Gillis et al., 2009*).

Phylogenetic analysis suggested that the GATA gene family had undergone distinct expansion that separated the GATA123 and GATA456 subfamilies, both of which were subsequently expanded. Our results differed from findings on the evolution of the GATA gene family in protostomes but agreed with those on the evolution of vertebrates. In protostomes, only the GATA456 subfamily appeared to have undergone expansion (*Gillis, Bowerman & Schneider, 2008*). By contrast, the GATA123 and GATA456 subfamilies both expanded in deuterostomes through the retention of duplicated GATA genes during multiple whole-genome duplications (*Dehal & Boore, 2005*). Our molecular phylogenetic analysis, together with the conserved syntenic paralogs (*Gillis et al., 2009*), provided evidence to support the expansion through genome duplication.

## Origin of GATA2 paralogs

Several molecular mechanisms, such as gene duplication, exon shuffling, gene fission and fusion, retrotranspositon, and mobile elements, have been proposed to understand the origin of new genes (*Long et al., 2003*). Gene duplication events, including single-gene duplication, segmental duplication, and genome duplication, are crucial to produce new genes (*Bailey et al., 2002*; *Samonte & Eichler, 2002*). In the present study, the results of chromosomal synteny analysis and gene-neighborhood synteny analysis indicated that the two GATA2 paralogs were generated through genome duplication in teleosts.

The fate of newborn genes is diverse. Some scholars believed that a number of a duplicate gene pairs eventually become nonfunctional and that most duplicates eventually perish as pseudogenes (*Bailey, Poulter & Stockwell, 1978*). Gene duplicates possibly acquire new functions (neofunctionalization) or undergo subfunctionalization and are preserved in a lineage (*Force et al., 1999*; *Kimura & King, 1979*; *Li, 1980*). During whole-genome duplication of yeast, arabidopsis, rice and tetraodon, all of the genes were duplicated, but only 10–30% of new genes were preserved, and others were lost in evolution (*Byrne & Wolfe, 2005*; *Paterson et al., 2006*). In the present study, ohnolog gone missing (ogm) was observed throughout the evolution of the GATA gene. Based on phylogenetic analysis, we conjectured that GATA1-ogm and GATA4-ogm occurred after 2R, which is consistent with a former study (*Gillis et al., 2009*). Most GATA paralogs, except GATA2, were lost after 3R in teleosts. GATA2 may have been preserved in the evolutionary process because of environmental pressure and further supports that the two GATA2 paralogs originated from 3R duplication.

## Structures of the GATA2a and GATA2b genes

The structure of GATA genes is generally conserved, as shown in protostomes and deuterostomes relative to vertebrate transcriptomes (*Gillis, Bowerman & Schneider, 2008*; *Gillis et al., 2009*). In the present study, we examined the conservation of the exon/intron structures of GATA2a and GATA2b in teleosts. The genomic structure of GATA2 was conserved; all GATA2 genes, except for *Pf* GATA2b, contained five exons in CDS. The lengths of the five exons were conserved, but the lengths of the introns varied. Introns are important indicators in eukaryotic evolution, where the gain and loss of introns reflect positive correlation or negative correlation with the coding-sequence evolution rate (*Carmel et al., 2007*; *Slamovits & Keeling, 2009*). The intron lengths of GATA2a were generally shorter than those of GATA2b, suggesting that the two GATA2 genes had

diverged. Meanwhile, motif prediction showed an additional motif 4 in GATA2b. This motif might separate GATA2b from GATA2a functionally, which was consistent with the phylogenetic results. These results implied that GATA2a and GATA2b in teleosts separated from each other and generated different structures and functions. We inferred that the sequence of GATA2a and GATA2b had been changed under selection pressure.

## Potential for functional divergence of GATA2a and GATA2b

In general, new genes evolve with rapid changes in their sequence and structure (*Wang et al., 2002*; *Zhang, Zhang & Rosenberg, 2002*), and mutation is the initial condition in evolution. Positive Darwinian selection may be another important force driving the evolution of new genes (*Ohta, 1994*; *Walsh, 1995*). The evolutionary rates of gene pairs that originated from duplication are usually different, and the rapid evolution of one of the gene pairs is a general phenomenon (*Johnson et al., 2001*; *Wang et al., 2002*). In the present study, the teleost GATA2 phylogenetic tree provided evidence that the evolutionary rate of GATA2b was faster than that of GATA2a under current environmental pressure. Thus, GATA2b has likely diverged from an ancestral GATA2 more similar to present GATA2a paralog.

The amino acid sequences of GATA genes contain the well-conserved N-terminal and C-terminal zinc finger motifs, which significantly contribute to structure and function. In the present study, the two zinc finger motifs were highly conserved in GATA2a and GATA2b in teleosts. Two mutated amino acid sites were found located in the two zinc finger motifs in GATA2b relative to GATA2a. Measuring the rate of relaxation and determining the presence of amino acid residue under positive selection are crucial to determine whether positive selection has driven the evolution of the GATA2 paralogs and whether or not selection constraints affect GATA2 genes after duplication in teleosts. The results of selection pressure analysis provided evidence of purifying selection, and one site (356P) in GATA2b was predicted to have undergone a strong positive selection. Interestingly, this site was located in the C-terminal zinc finger motif, which has been inferred to play an important role during evolution. Therefore, this positively selected site might affect the binding activity of GATA2b or even affect the selection of binding sites, resulting in the functional divergence between GATA2a and GATA2b.

In the current study, transcriptional analysis was performed using qRT-PCR. We focused on the overall expression pattern but not the individual differences, so pooled samples were used. Three experimental repeats for each pooled sample were performed to ensure the operational accuracy and the results could effectively reflect the actual expression levels on average. The expression patterns of GATA2a and GATA2b were similar in most somatic organs, but sexual dimorphic expression was apparent, especially in the spleen and the gonad. Previous studies have shown that new genes have evolved in conjunction with rapid changes in expression (*Wang et al., 2002*; *Zhang, Zhang & Rosenberg, 2002*), and the differential expression of these genes was believed to be the first step in functional divergence. The classical model for the evolution of duplicate genes identifies two possibilities: one is that one of the duplicated genes degenerates by accumulating deleterious mutations; the other is that one duplicate acquires a new adaptive function (*Ohno, 1970*). However, the duplication–degeneration–complementation (DDC)

model predicts that the duplicate gene preservation involves the partitioning of ancestral functions rather than the evolution of new functions (*Force et al., 1999*). Moreover, the expression levels of GATA2b in the brain, the pituitary gland, and the gonad differed between females and males in tilapia (*Zhang, 2009*). Based on the results of our present study, we hypothesize that the differential transcription of the GATA2 paralogs in tongue sole follow the DDC model; that is, GATA2a and GATA2b partitioned the ancestral functions of GATA2 in teleosts. GATA2a might have maintained the functions of GATA2 in hemopoiesis and in the multiplication and differentiation of hematopoietic stem cells, whereas GATA2b might have acquired some functions related to sexual differentiation and gonad development or sexual maturation. These results provide preliminary evidence that the duplicated GATA2 genes may have undergone neofunctionalization in teleosts.

## CONCLUSIONS

In summary, we investigate the origin of teleost GATA2a/b genes and reports for the first time that two GATA2 genes are present in teleosts as a result of TGD. In addition, our results indicate possible neofunctionalization of the duplicated GATA2 genes, providing novel insight into the teleost GATA gene family and future functional studies of GATA2 in fish.

### Funding

This work was supported by the National High-Tech Research and Development Program of China (2012AA10A408 and 2012AA10A402). The funders had no role in study design, data collection and analysis, decision to publish, or preparation of the manuscript.

### Grant Disclosures

The following grant information was disclosed by the authors:
National High-Tech Research and Development Program of China: 2012AA10A408, 2012AA10A402.

### Competing Interests

The authors declare there are no competing interests.

### Author Contributions

- Jinxiang Liu performed the experiments, analyzed the data, wrote the paper, prepared figures and/or tables, reviewed drafts of the paper.
- Jiajun Jiang prepared figures and/or tables, reviewed drafts of the paper.
- Zhongkai Wang contributed reagents/materials/analysis tools, reviewed drafts of the paper.
- Yan He analyzed the data, contributed reagents/materials/analysis tools, reviewed drafts of the paper.
- Quanqi Zhang conceived and designed the experiments, reviewed drafts of the paper.

## Animal Ethics

The following information was supplied relating to ethical approvals (i.e., approving body and any reference numbers):

All research was conducted in accordance with the Institutional Animal Care and Use Committee of the Ocean University of China and with the China Government Principles for the Utilization and Care of Vertebrate Animals Used in Testing, Research, and Training (State science and technology commission of the People's Republic of China for No. 2, October 31,1988: http://www.gov.cn/gongbao/content/2011/content_1860757.htm).

## Data Availability

Raw data is uploaded as Data S1.

## Supplemental Information

Supplemental information for this article can be found online at http://dx.doi.org/10.7717/peerj.1790#supplemental-information.

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
