# Peer review of "Origin and evolution of GATA2a and GATA2b in teleosts: insights from tongue sole, Cynoglossus semilaevis"

_PeerJ, doi:10.7717/peerj.1790_

## Round 0.1 · original submission · Major Revisions

· Academic Editor

Major Revisions

While this manuscript contains interesting results it requries substantial revision. In particular:
- please identify your research question and then link it to methods, results and conclusions
- please make sure that all your statements are substantiated - do not make claims not supported by your results
- please address all the suggested changes and comments from the reviewers, in particular those in Experimental Design and Validity of Findings sections
- please improve your writing style, including correction of all grammatical errors

Reviewer 1 ·

Basic reporting

In my opinion, the major weakness of this manuscript lies in its English articulation which currently falls short of what most journals would consider a ‘professional standard’ of expression. There are many grammatical and sentence structure issues in conjunction with poor or inappropriate word choices which make the rationale, results, and subsequent conclusions of the study hard to follow. I have provided some editorial suggestions for improving the abstract in an attached file as an example; however, further editing is beyond the scope of a reviewer and I strongly advise the authors to enlist scientific editorial support to better communicate these findings to the scientific community. Whether or not such editorial support falls within the scope of PeerJ, or to what degree of improvement (if any) is required for publication in PeerJ, I leave at the discretion of the handling editor.

Experimental design

As stated above for Basic Reporting, clear communication is lacking. A research question was not clearly defined, the knowledge gap to be investigated could be loosely inferred but was not clearly identified, nor was a statement provided as to how the study contributes to filling that knowledge gap (all of which should have been made clear in the Introduction). Because such rationale was not clearly presented I have a hard time judging the completeness or appropriateness of the methodology. Nevertheless, based on general inference, I see few flaws in the methodology apart from the use of a single reference gene in qPCR analysis. I would prefer the authors use at least 2 references or provide evidence for the stability of their single gene (i.e. based on a previous similar study which demonstrated stability or selection from a list of multiple references based on stability factor estimates). Also, please indicate how many female and male tongue sole you used to generate the qPCR comparisons and provide statistical comparisons for relative expression between sexes for each tissue tested (i.e. t-test or non-parametric equivalent as appropriate)

Validity of the findings

Most of the findings in this manuscript appear scientifically valid; however, how the conclusions connect to the original research question and its scientific importance are not clearly presented which makes interpretation of the findings significance difficult.

The authors state their analyses “revealed evidence of neofunctionalization of the duplicated GATA2 genes”. I think this is overstated, as the authors provide no evidence that proteins translated by these genes actually carry out discrete functions. At best this data is suggestive of neofunctionalization and discussion of this fact should be treated as speculation.

As stated above, please provide evidence for reference gene stability between male and females to validate the qPCR relative expression comparisons provided for each tissue. Also, if comparisons are to be made regarding relative expression between tissues (as currently presented in L249-251), further evidence is required to demonstrate the reference gene(s) is equally expressed in all tissue types being compared. Lastly, a measure for statistical significance (i.e. p-value) between relative expression of male and female sole for each tissue is needed, otherwise you cannot state as you have done in Line 251 that GATA2a was higher in female spleen than male spleen. Statistically significant differences should be indicated on Figure 6.

Additional comments

Figure 2 – The proline site is indicated as having undergone positive selection. Positive selection relative to what?

Phylogenetic Figures 1 and 3 – Please define the numbers presented on each major branch.

Annotated reviews are not available for download in order to protect the identity of reviewers who chose to remain anonymous.

·

Basic reporting

The issues in this section are minor, that the authors can revise:
1) Grammar issues – I will not list these exhaustively, but the authors need to go through the manuscript correcting grammatical issues. (e.g L40-41)
2) L173 – “…showed that GATA123 genes evolved more rapidly than their GATA456 counterparts…” Please be specific and describe how this is shown.
3) L190 – “Compared with GATA2a, GATA2b was more active under the current environmental pressure and evolved more rapidly.” This sentence is more appropriate in discussion, but it is also not very clear of what the ‘current environmental pressure’ is? Please clarify.
4) L200 define gene differentiation
5) L201 and throughout, remove the word ‘proved’ ‘ (L299; L296)’ and ‘proven’ These terms are generally not used, but rather ‘shown’ ‘provide evidence for’ ‘confirmed’ etc.
6) L67 Please provide a citation for the cause—effect relationship between gene copy number and species diversity
7) L87 Provide the method of sacrificing (killing) the fish
8) L75- suggest change “…to illustrate the origin and evolution of teleosts…” to “…to illustrate the origin and evolution of these genes in teleosts…”
9) L96 change wording, i.e. “sequenced and available from the Ensembl database”
10) L98 please provide one or two sentences on the construction of the transcriptome and the statistics of the transcriptome (e.g. completeness etc).
11) L100 please provide accession for the tongue sole genome
12) L303 ‘Intron is a symbol…’ – please reword for clarity
13) L318 last sentence of paragraph does not make sense, please clarify ‘…was supposed to be…’
14) L347-348 “Moreover, the expression levels of GATA2b in the brain, the gland, and the gonad differed between females and males in tilapia.” Please cite, and what gland is being referred to?
15) Figure 3 caption: there are species acronyms that are not defined in the caption.
L105 The authors use an ‘unpublished Japanese flounder genome’ – but without it being published the work is not reproducible; a way around this would be to upload the gene specific data (just that specifically needed to reproduce this study) to a server. It is possible that this information is provided in the supplemental files, if so, please refer to it in the manuscript specifically.

Experimental design

16) L142-150 Additional information is needed within the RT-qPCR methods section. Specifically, there is no statement regarding the standard curve (which is essential for relative RT-qPCR), except at L144 “cDNAs from at least three individuals were mixed up to 10 ng/ul as templates”. It is not clear whether this is for the standard curve or for actual sample assessment. Efficiency values of primers are given by the standard curve, and these need to be presented. Efficiency curves should encompass the range of the test samples. Furthermore, regarding the pooling, there is a lack of clarity regarding exactly what the sample number is for the different tissues (and genes); if individuals are pooled, this needs to be explicitly stated. Other missing information for RT-qPCR: size of the amplicons; specifics on what negative controls are used (-RT, NTC); and the melt curve starting temperature. These are all important to report in order for the reader to evaluate the work.
17) L251 The description of the methods for differential expression analysis are essentially absent. The only thing stated is the 2^ddCt is used for normalization. Problematically, there is discussion of ‘statistical difference between sexes’ without any mention of how statistics were performed, or presentation of p-values.

Validity of the findings

18) L246 and L334 The authors describe differences between the levels of the two genes. Comparing expression levels between genes, even when both genes are normalized by the same normalizer gene, cannot be done with relative qPCR. This is the purpose of absolute quantification qPCR. Relative expression values can only be compared between conditions (or tissues in this case) within a gene, not between genes. Even if efficiencies are approximately equal, primer binding efficiency can lead to differences in the normalized expression levels between two genes. Therefore, using the data present in the paper, the authors either need to remove this statement about the differences in expression levels between the gene (e.g. gene x is expressed much higher than gene y), or at the very least state this is just some evidence towards this, but it not definitive, and that absolute quantification would be needed to validate this evidence. For more detail on this, please see the following article, in particular Figure 2: Boyle, B., N. Dallaire, and J. MacKay, 2009 Evaluation of the impact of single nucleotide polymorphisms and primer mismatches on quantitative PCR. BMC Biotechnology 9: 75.

Additional comments

Here, Liu et al. analyze GATA2a and GATA2b in teleosts. The authors provide evidence that these two genes present in teleosts are a result of the teleost-specific WGD. Further, the authors make suggestions as to which is the ancestral form of the gene, and that the two genes may have different functional roles. The authors find good evidence of these genes differing in specific important amino acids (Figure 2), and that they arose from the teleost WGD (Figure 4). The description of the work is generally concise and clear (except for some areas highlighted below).
My main criticisms are regarding the RT-qPCR analysis
The most important criticisms for the authors to deal with are those found in the sections of this review "Experimental Design", and "Validity of the Findings" (specifically points 16-18).

---

## Round 0.2 · Minor Revisions

· Academic Editor

Minor Revisions

Please make all the corrections suggested by the reviewers. Please clarify the sample size for gene expression and pooling if any.

Please clarify how "random dissection" was done. I'm assuming you mean that the fish were selected randomly but I doubt that your selection was truly random (in a statistical sense) as that would mean using procedures (please describe procedures used if you really used random selection procedures) which give each fish the same chance to be selected. If you haven't used random selection procedures please deleted the word "randomly". If you did please change to "randomly selected and then dissected".

Reviewer 1 ·

Basic reporting

The manuscript improved following revision, however I still think effective communication is one of the major shortcomings of this work. Nevertheless, provided the authors include or address the suggested alterations which I have provided as a annotated attachment, I feel the reporting will be sufficiently accurate and reproducible to recommend publication.

Experimental design

I see no major weaknesses in the study design currently presented.

Validity of the findings

In revision, the authors have provided a statistical analysis (ANOVA) for assessing differential expression of GATA2a/b between male and female fish (summarized in figure 6). Unfortunately gene expression analysis of this kind is usually non-normally distributed and the authors do not have a sufficient sample size to confirm normality of their data. Therefore the authors need to either: A) normalize the data using a log-transform function prior to running their ANOVA (a practice commonly applied to gene expression analyses and generally accepted as sufficient for normalizing this type of data), or B) use a non-parametric statistical test such as a Kruskal–Wallis test or similar instead of the ANOVA. Further, the authors described significant differences between levels of GATA2a/b between tissues (Ln 277-278 in the annotated document attached), but no results are presented showing which comparisons had p-values <0.05. This could be added to Figure six but it may result in the figure becoming too cluttered. I suggest adding another figure or supplemental figure showing significant differences observed between tissues.

Lastly, although word choice was improved in some instances, I feel the revised manuscripts still overstates the findings regarding functional divergence. As stated previously, you have not actually tested the function of these genes. Thus, the gene expression differences you observed could simply be explained though transcriptional stimulation of the two genes with different promoters or regulatory factors and still have the exact same function. As you did not do any functional protein tests in this study you have no proof to say otherwise, only speculation. Please consider my recommendations regarding such wording in the annotated attachment.

Additional comments

Please remove the word "level" from the y-axis of figure 6. I also suggest adding "GATA2a" and "GATA2b" as a second x-axis label to figure 6A and 6B, respectively, for visual clarity.

Also, although the figure legends have been appropriately revised in the manuscript file, the legends associated with each figure image were not updated in my reviewing document. Please check and ensure the updated legends are appropriately assigned to the figures prior to final publication.

Annotated reviews are not available for download in order to protect the identity of reviewers who chose to remain anonymous.

·

Basic reporting

2) L74 grammatical issue: “In teleost, the research on the two GATA2 gene is few.”

3) L326-329 “In the present study, the teleost GATA2 phylogenetic tree provided evidence that the evolutionary rate of GATA2b was faster than that of GATA2a under the current environmental pressure, including temperature, illumination, and dissolved oxygen.” I do not think the authors should state the specifics of the environmental pressure unless they have evidence for it. Removing "...including temperature... etc" would be more correct - the authors corrected the issue I had with this sentence in my first review by stating that the evidence was from the phylogenetic tree.

4) L346 One instance of the word ‘proven’ remains in the document, please correct (as requested, and mostly corrected, in my first review).

5) L357 Stating ‘the gland’ of the fish is not specific enough, it is still unclear what gland the authors are referring to. Is it the pituitary gland, the thyroid gland, etc.?

6) L312 in revised ms (pdf): “Intron is a symbol in eukaryotic evolution, and the gain and loss of intron reflect positive correlation or negative correlation with the coding-sequence evolution rate. (Carmel et al. 2007; Slamovits & Keeling 2009).” Needs to be revised for grammar, and remove period prior to citations. For example: “Introns are important indicators in eukaryotic evolution, where the gain and loss of introns reflect …” or similar.

7) L160-161 “…and the melt curve starting temperatures were 82°C, 78°C, and 83°C.” This is probably not the starting temperatures of the melt curve assessment, but rather the temperatures at which the amplicons melted. The starting temperatures would be around ~55 degrees or so (i.e. the primer annealing temperature), then increase incrementally up to 90 degrees or so. This needs to be corrected for clarity.

8) L299 “During whole-genome duplication, all of the genes were duplicated, but only 10%–30% of new genes were preserved, and others were lost in evolution (Byrne & Wolfe 2005; Paterson et al. 2006).” For clarity, please be specific as to what species for which this was observed.

Experimental design

1) As I stated in my first review (comment 16 within ‘Experimental Design’), the sample size for gene expression work remains very unclear. The following is from the revised manuscript:

L92-94: Healthy tongue sole (three females and males) of one-year-old were dissected randomly. The fish were anesthetized and then killed by breaking verterbra. Brain, heart, intestine, kidney, liver, spleen, and gonad tissues were collected in triplicate.

This suggests to me that the sample size is n = 3 per sex per tissue. But then, in the revised manuscript:
L154-155: “cDNAs from three individuals were pooled as templates (10 ng/μL) for sample assessment.”
It appears therefore that the sample size is n = 1 for each tissue (as three individuals were pooled into one)? Once three individuals are pooled into one, this must be treated as a single individual (n = 1). If this is the case, it is not clear how the standard deviations are presented in Figure 6, and this must be corrected. If there is no biological replication, there can be no standard deviation (or standard error of the mean). Furthermore, if the sample size is 1, this needs to be stated as such in the manuscript, for example as preliminary results. I do not see how statistics could have been done on an n = 1 sample size (e.g. t-test as is stated in the text). It is possible that the authors meant to say ‘diluted’ rather than pooled, but without some statement of sample sizes per sex per tissue, it is impossible for me to determine. This is necessary to clarify, correct, or temper the conclusion as described above before the gene expression experiment's results can be assessed by the reader.

Validity of the findings

No Comments

Additional comments

Liu et al. have corrected most of the main issues highlighted in my first review. There remain only a couple areas that need revision. The issue on sample size highlighted in 'Experimental Design' especially requires some consideration and revision.

---

## Round 0.3 · Minor Revisions

· Academic Editor

Minor Revisions

I am still unclear about the replication - this has been questioned by the reviewers as well. The new sentence "Three biological replicates were used for each tissue, including no-template controls and each replicate consisted of a sample pool of three tissues." is not really explaining replication. Please explain clearly if:

- the three biological replicates were from different individuals

- the pooling involved samples from the same individual or different individuals and if the pooled samples were from the same organ (:three tissues" would usually mean three types of tissues so it sounds like you pooled samples from different organs? If you pooled samples of the same organ from the same individual or samples of the same organs from different individuals please clarify)

- 3 females ad 3 males or 3 individuals in total were used

It would be much better to replace "tissues" by "organs" as heart or brain or kidney are organs, please see definition of organ and tissue.

Please correct any typographical errors, for example "vertebra"is misspelled in line 101 and I suspect that more than one was broken so plural should have been used. Don't you really mean that you severed the spinal cord of the fish?

---

## Round 0.4 · Minor Revisions

· Academic Editor

Minor Revisions

I have requested reasonably minor changes and while the authors have replied to some (but not all) of my comments I cannot see the changes made in the manuscript. Please make the changes and reply to all comments:

I am still unclear about the replication - this has been questioned by the reviewers as well. The new sentence "Three biological replicates were used for each tissue, including no-template controls and each replicate consisted of a sample pool of three tissues." is not really explaining replication. Please explain clearly if:

- the three biological replicates were from different individuals - this is authors reply but not clarified in the text

- the pooling involved samples from the same individual or different individuals and if the pooled samples were from the same organ (:three tissues" would usually mean three types of tissues so it sounds like you pooled samples from different organs? If you pooled samples of the same organ from the same individual or samples of the same organs from different individuals please clarify) - this is authors reply but I cannot see it in the manuscript

- 3 females ad 3 males or 3 individuals in total were used

There are no replies to the other comments:
It would be much better to replace "tissues" by "organs" as heart or brain or kidney are organs, please see definition of organ and tissue.

Please correct any typographical errors, for example "vertebra"is misspelled in line 101 and I suspect that more than one was broken so plural should have been used. Don't you really mean that you severed the spinal cord of the fish? - I can see that this last change was made.

---

## Round 0.5 · Minor Revisions

· Academic Editor

Minor Revisions

Dear authors,

thank you for your letter and really sorry for the delay. I can see the changes in the manuscript. I would like you to include a brief comment in Discussion about the use of technical versus biological replicates. The fact that you pooled organs from the individuals and then took samples from the pool has implications for your conclusions.

Once again I sincerely apologise for the delay and promise to be much faster next time.

---

## Round 0.6 · accepted · Accept

· Academic Editor

Accept

Thank you for making the change, The manuscript is now accepted for publication in PeerJ.